# META-LEARNING UPDATE RULES FOR UNSUPERVISED REPRESENTATION LEARNING

**Luke Metz**
Google Brain
lmetz@google.com

**Niru Maheswaranathan**
Google Brain
nirum@google.com

**Brian Cheung**
University of California, Berkeley
bcheung@berkeley.edu

**Jascha Sohl-Dickstein**
Google Brain
jaschasd@google.com

## ABSTRACT

A major goal of unsupervised learning is to discover data representations that are useful for subsequent tasks, without access to supervised labels during training. Typically, this involves minimizing a surrogate objective, such as the negative log likelihood of a generative model, with the hope that representations useful for subsequent tasks will arise as a side effect. In this work, we propose instead to directly target later desired tasks by meta-learning an unsupervised learning rule which leads to representations useful for those tasks. Specifically, we target semi-supervised classification performance, and we meta-learn an algorithm – an unsupervised weight update rule – that produces representations useful for this task. Additionally, we constrain our unsupervised update rule to a be a biologically-motivated, neuron-local function, which enables it to generalize to different neural network architectures, datasets, and data modalities. We show that the meta-learned update rule produces useful features and sometimes outperforms existing unsupervised learning techniques. We further show that the meta-learned unsupervised update rule generalizes to train networks with different widths, depths, and nonlinearities. It also generalizes to train on data with randomly permuted input dimensions and even generalizes from image datasets to a text task.

## 1 INTRODUCTION

Supervised learning has proven extremely effective for many problems where large amounts of labeled training data are available. There is a common hope that unsupervised learning will prove similarly powerful in situations where labels are expensive, impractical to collect, or where the prediction target is unknown during training. Unsupervised learning however has yet to fulfill this promise. One explanation for this failure is that unsupervised representation learning algorithms are typically mismatched to the target task. Ideally, learned representations should linearly expose high level attributes of data (e.g. object identity) and perform well in semi-supervised settings. Many current unsupervised objectives, however, optimize for objectives such as log-likelihood of a generative model or reconstruction error, producing useful representations only as a side effect.

Unsupervised representation learning seems uniquely suited for meta-learning (Hochreiter et al., 2001; Schmidhuber, 1995). Unlike most tasks where meta-learning is applied, unsupervised learning does not define an explicit objective, which makes it impossible to phrase the task as a standard optimization problem. It is possible, however, to directly express a *meta-objective* that captures the quality of representations produced by an unsupervised update rule by evaluating the usefulness of the representation for candidate tasks. In this work, we propose to *meta-learn* an unsupervised update rule by meta-training on a meta-objective that directly optimizes the utility of the unsupervised representation. Unlike hand-designed unsupervised learning rules, this meta-objective directly targets the usefulness of a representation generated from unlabeled data for later supervised tasks.

By recasting unsupervised representation learning as meta-learning, we treat the creation of the unsupervised update rule as a transfer learning problem. Instead of learning transferable features,

we learn a transferable learning rule which does not require access to labels and generalizes across both data domains and neural network architectures. Although we focus on the meta-objective of semi-supervised classification here, in principle a learning rule could be optimized to generate representations for *any* subsequent task.

## 2 RELATED WORK

### 2.1 UNSUPERVISED REPRESENTATION LEARNING

Unsupervised learning is a topic of broad and diverse interest. Here we briefly review several techniques that can lead to a useful latent representation of a dataset. In contrast to our work, each method imposes a manually defined training algorithm or loss function whereas we *learn* the algorithm that creates useful representations as determined by a meta-objective.

Autoencoders (Hinton and Salakhutdinov, 2006) work by first compressing and optimizing reconstruction loss. Extensions have been made to de-noise data (Vincent et al., 2008; 2010), as well as compress information in an information theoretic way (Kingma and Welling, 2013). Le et al. (2011) further explored scaling up these unsupervised methods to large image datasets.

Generative adversarial networks (Goodfellow et al., 2014) take another approach to unsupervised feature learning. Instead of a loss function, an explicit min-max optimization is defined to learn a generative model of a data distribution. Recent work has shown that this training procedure can learn unsupervised features useful for few shot learning (Radford et al., 2015; Donahue et al., 2016; Dumoulin et al., 2016).

Other techniques rely on self-supervision where labels are easily generated to create a non-trivial 'supervised' loss. Domain knowledge of the input is often necessary to define these losses. Noroozi and Favaro (2016) use unscrambling jigsaw-like crops of an image. Techniques used by Misra et al. (2016) and Sermanet et al. (2017) rely on using temporal ordering from videos.

Another approach to unsupervised learning relies on feature space design such as clustering. Coates and Ng (2012) showed that k-means can be used for feature learning. Xie et al. (2016) jointly learn features and cluster assignments. Bojanowski and Joulin (2017) develop a scalable technique to cluster by predicting noise. Other techniques such as Schmidhuber (1992), Hochreiter and Schmidhuber (1999), and Olshausen and Field (1997) define various desirable properties about the latent representation of the input, such as predictability, complexity of encoding mapping, independence, or sparsity, and optimize to achieve these properties.

### 2.2 META LEARNING

Most meta-learning algorithms consist of two levels of learning, or 'loops' of computation: an *inner loop*, where some form of learning occurs (e.g. an optimization process), and an *outer loop* or *meta-training* loop, which optimizes some aspect of the inner loop, parameterized by *meta-parameters*. The performance of the inner loop computation for a given set of meta-parameters is quantified by a *meta-objective*. *Meta-training* is then the process of adjusting the meta-parameters so that the inner loop performs well on this meta-objective. Meta-learning approaches differ by the computation performed in the inner loop, the domain, the choice of meta-parameters, and the method of optimizing the outer loop.

Some of the earliest work in meta-learning includes work by Schmidhuber (1987), which explores a variety of meta-learning and self-referential algorithms. Similarly to our algorithm, Bengio et al. (1990; 1992) propose to learn a neuron local learning rule, though their approach differs in task and problem formulation. Additionally, Runarsson and Jonsson (2000) meta-learn supervised learning rules which mix local and global network information. A number of papers propose meta-learning for few shot learning (Vinyals et al., 2016; Ravi and Larochelle, 2016; Mishra et al., 2017; Finn et al., 2017; Snell et al., 2017), though these do not take advantage of unlabeled data. Others make use of both labeled and unlabeld data (Ren et al., 2018). Hsu et al. (2018) uses a task created with no supervision to then train few-shot detectors. Garg (2018) use meta-learning for unsupervised learning, primarily in the context of clustering and with a small number of meta-parameters.

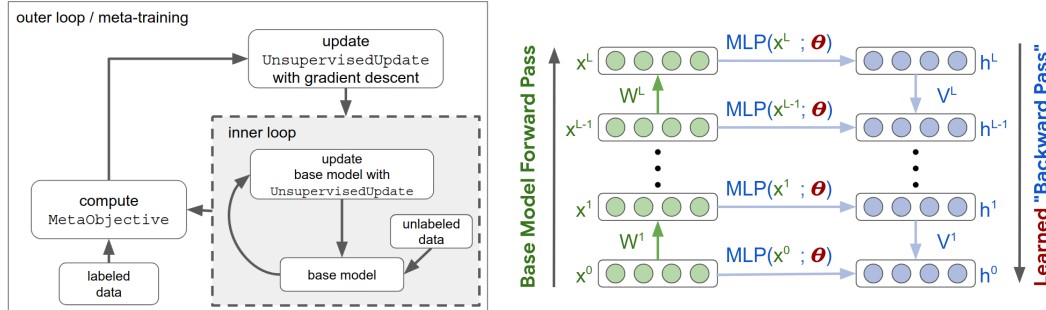

Figure 1: **Left:** Schematic for meta-learning an unsupervised learning algorithm. The inner loop computation consists of iteratively applying the UnsupervisedUpdate to a base model. During meta-training the UnsupervisedUpdate (parameterized by $\theta$) is itself updated by gradient descent on the MetaObjective. **Right:** Schematic of the base model and UnsupervisedUpdate. Unlabeled input data, $x_0$, is passed through the base model, which is parameterised by $W$ and colored green. The goal of the UnsupervisedUpdate is to modify $W$ to achieve a top layer representation $x^L$ which performs well at few-shot learning. In order to train the base model, information is propagated backwards by the UnsupervisedUpdate in a manner analogous to backprop. Unlike in backprop however, the backward weights $V$ are decoupled from the forward weights $W$. Additionally, unlike backprop, there is no explicit error signal as there is no loss. Instead at each layer, and for each neuron, a learning signal is injected by a meta-learned MLP parameterized by $\theta$, with hidden state $h$. Weight updates are again analogous to those in backprop, and depend on the hidden state of the pre- and post-synaptic neurons for each weight.

To allow easy comparison against other existing approaches, we present a more extensive survey of previous work in meta-learning in table form in Table 1, highlighting differences in choice of task, structure of the meta-learning problem, choice of meta-architecture, and choice of domain.

To our knowledge, we are the first meta-learning approach to tackle the problem of unsupervised representation learning, where the inner loop consists of unsupervised learning. This contrasts with transfer learning, where a neural network is instead trained on a similar dataset, and then fine tuned or otherwise post-processed on the target dataset. We additionally believe we are the first representation meta-learning approach to generalize across input data modalities as well as datasets, the first to generalize across permutation of the input dimensions, and the first to generalize across neural network architectures (e.g. layer width, network depth, activation function).

## 3    MODEL DESIGN

We consider a multilayer perceptron (MLP) with parameters $\phi_t$ as the *base model*. The inner loop of our meta-learning process trains this base model via iterative application of our learned update rule. See Figure 1 for a schematic illustration and Appendix A for a more detailed diagram.

In standard supervised learning, the 'learned' optimizer is stochastic gradient descent (SGD). A supervised loss $l(x, y)$ is associated with this model, where $x$ is a minibatch of inputs, and $y$ are the corresponding labels. The parameters $\phi_t$ of the base model are then updated iteratively by performing SGD using the gradient $\frac{\partial l(x,y)}{\partial \phi_t}$. This supervised update rule can be written as $\phi_{t+1} = \text{SupervisedUpdate}(\phi_t, x_t, y_t; \theta)$, where $t$ denotes the inner-loop iteration or step. Here $\theta$ are the meta-parameters of the optimizer, which consist of hyper-parameters such as learning rate and momentum.

In this work, our learned update is a parametric function which does not depend on label information, $\phi_{t+1} = \text{UnsupervisedUpdate}(\phi_t, x_t; \theta)$. This form of the update rule is general, it encompasses many unsupervised learning algorithms and all methods in Section 2.1.

In traditional learning algorithms, expert knowledge or a simple hyper-parameter search determines $\theta$, which consists of a handful of meta-parameters such as learning rate and regularization constants. In contrast, our update rule will have orders of magnitude more meta-parameters, including the

| Method | Inner loop updates | Outer loop updates, meta- | | | Generalizes to |
|---|---|---|---|---|---|
| | | parameters | objective | optimizer | |
| Hyper parameter optimization Jones (2001); Snoek et al. (2012); Bergstra et al. (2011); Bergstra and Bengio (2012) | many steps of optimization | optimization hyper-parameters | training or validation set loss | Baysian methods, random search, etc | test data from a fixed dataset |
| Neural architecture search Stanley and Miikkulainen (2002); Zoph and Le (2017); Baker et al. (2017); Zoph et al. (2018); Real et al. (2017) | supervised SGD training using meta-learned architecture | architecture | validation set loss | RL or evolution | test loss within similar datasets |
| Task-specific optimizer (eg for quadratic function identification) (Hochreiter et al., 2001) | adjustment of model weights by an LSTM | LSTM weights | task loss | SGD | similar domain tasks |
| Learned optimizers Jones (2001); Maclaurin et al. (2015); Andrychowicz et al. (2016); Chen et al. (2016); Li and Malik (2017); Wichrowska et al. (2017); Bello et al. (2017) | many steps of optimization of a fixed loss function | parametric optimizer | average or final loss | SGD or RL | new loss functions (mixed success) |
| Prototypical networks Snell et al. (2017) | apply a feature extractor to a batch of data and use soft nearest neighbors to compute class probabilities | weights of the feature extractor | few shot performance | SGD | new image classes within similar dataset |
| MAML Finn et al. (2017) | one step of SGD on training loss starting from a meta-learned network | initial weights of neural network | reward or training loss | SGD | new goals, similar task regimes with same input domain |
| Evolved Policy Gradient Houthooft et al. (2018) | performing gradient descent on a learned loss | parameters of a learned loss function | reward | Evolutionary Strategies | new environment configurations, both in and not in meta-training distribution. |
| Few shot learning (Vinyals et al., 2016; Ravi and Larochelle, 2016; Mishra et al., 2017) | application of a recurrent model, e.g. LSTM, Wavenet. | recurrent model weights | test loss on training tasks | SGD | new image classes within similar dataset. |
| Meta-unsupervised learning for clustering Garg (2018) | run clustering algorithm or evaluate binary similarity function | clustering algorithm + hy-perparameters, binary similarity function | empirical risk mini-mization | varied | new clustering or similarity measurement tasks |
| Learning synaptic learning rules (Bengio et al., 1990; 1992) | run a synapse-local learning rule | parametric learning rule | supervised loss, or similarity to biologically-motivated network | gradient descent, simulated annealing, genetic algorithms | similar domain tasks |
| **Our work — metalearning for unsupervised representation learning** | **many applications of an unsupervised update rule** | **parametric update rule** | **few shot classifica-tion after unsuper-vised pre-training** | **SGD** | **new base models (width, depth, nonlinearity), new datasets, new data modalities** |

Table 1: A comparison of published meta-learning approaches.

weights of a neural network. We train these meta-parameters by performing SGD on the sum of the MetaObjective over the course of (inner loop) training in order to find optimal parameters $\theta^*$,

$$\theta^* = \underset{\boldsymbol{\theta}}{\operatorname{argmin}} \, \mathbb{E}_{\text{task}} \left[ \sum_t \text{MetaObjective}(\phi_t) \right], \tag{1}$$

that minimize the meta-objective over a distribution of training tasks. Note that $\phi_t$ is a function of $\theta$ since $\theta$ affects the optimization trajectory.

In the following sections, we briefly review the main components of this model: the base model, the UnsupervisedUpdate, and the MetaObjective. See the Appendix for a complete specification. Additionally, code and meta-trained parameters $\theta$ for our meta-learned UnsupervisedUpdate is available[1].

### 3.1 BASE MODEL

Our base model consists of a standard fully connected multi-layer perceptron (MLP), with batch normalization (Ioffe and Szegedy, 2015), and ReLU nonlinearities. We chose this as opposed to a convolutional model to limit the inductive bias of convolutions in favor of learned behavior from the UnsupervisedUpdate. We call the pre-nonlinearity activations $z^1, \cdots, z^L$, and post-nonlinearity activations $x^0, \cdots, x^L$, where $L$ is the total number of layers, and $x^0 \equiv x$ is the network input (raw data). The parameters are $\phi = \left\{ W^1, b^1, V^1, \cdots, W^L, b^L, V^L \right\}$, where $W^l$ and $b^l$ are the weights and biases (applied after batch norm) for layer $l$, and $V^l$ are the corresponding weights used in the backward pass.

### 3.2 LEARNED UPDATE RULE

We wish for our update rule to generalize across architectures with different widths, depths, or even network topologies. To achieve this, we design our update rule to be *neuron-local*, so that updates are a function of pre- and post- synaptic neurons in the base model, and are defined for any base model architecture. This has the added benefit that it makes the weight updates more similar to synaptic updates in biological neurons, which depend almost exclusively on local pre- and post-synaptic neuronal activity (Whittington and Bogacz, 2017). In practice, we relax this constraint and incorporate some cross neuron information to decorrelate neurons (see Appendix G.5 for more information).

To build these updates, each neuron $i$ in every layer $l$ in the base model has an MLP, referred to as an update network, associated with it, with output $h_b^l i = \text{MLP}\left( x_b^l i, z_b i^l, V^{l+1}, \delta^{l+1}; \theta \right)$ where $b$ indexes the training minibatch. The inputs to the MLP are the feedforward activations ($x^l$ & $z^l$) defined above, and feedback weights and an error signal ($V^l$ and $\delta^l$, respectively) which are defined below.

All update networks share meta-parameters $\theta$. Evaluating the statistics of unit activation over a batch of data has proven helpful in supervised learning (Ioffe and Szegedy, 2015). It has similarly proven helpful in hand-designed unsupervised learning rules, such as sparse coding and clustering. We therefore allow $h_{bi}^l$ to accumulate statistics across examples in each training minibatch.

During an unsupervised training step, the base model is first run in a standard feed-forward fashion, populating $x_{bi}^l, z_{bi}^l$. As in supervised learning, an error signal $\delta_{bi}^l$ is then propagated backwards through the network. Unlike in supervised backprop, however, this error signal is generated by the corresponding update network for each unit. It is read out by linear projection of the per-neuron hidden state $h$, $\delta_{bi}^l = \text{lin}\left(h_{bi}^l\right)$, and propagated backward using a set of learned 'backward weights' $(V^l)^T$, rather than the transpose of the forward weights $(W^l)^T$ as would be the case in backprop (diagrammed in Figure 1). This is done to be more biologically plausible (Lillicrap et al., 2016).

Again as in supervised learning, the weight updates ($\Delta W^l$) are a product of pre- and post-synaptic signals. Unlike in supervised learning however, these signals are generated using the per-neuron update

---

[1] https://github.com/tensorflow/models/tree/master/research/learning_unsupervised_learning

networks: $\Delta W_{ij}^l = \text{func}\left(h_{bi}^l, h_{bj}^{l-1}, W_{ij}\right)$. The full weight update (which involves normalization and decorrelation across neurons) is defined in Appendix G.5.

## 3.3 META-OBJECTIVE

The meta-objective determines the quality of the unsupervised representations. In order to meta-train via SGD, this loss must be differentiable. The meta-objective we use in this work is based on fitting a linear regression to labeled examples with a small number of data points. In order to encourage the learning of features that generalize well, we estimate the linear regression weights on one minibatch $\{x_a, y_a\}$ of $K$ data points, and evaluate the classification performance on a second minibatch $\{x_b, y_b\}$ also with $K$ datapoints,

$$\hat{v} = \underset{v}{\text{argmin}}\left(\left\|y_a - v^T x_a^L\right\|^2 + \lambda \left\|v\right\|^2\right), \qquad \text{MetaObjective}(\cdot; \phi) = \text{CosDist}\left(y_b, \hat{v}^T x_b^L\right), \quad (2)$$

where $x_a^L$, $x_b^L$ are features extracted from the base model on data $x_a$, $x_b$, respectively. The target labels $y_a$, $y_b$ consist of one hot encoded labels and potentially also regression targets from data augmentation (e.g. rotation angle, see Section 4.2). We found that using a cosine distance, $\text{CosDist}$, rather than unnormalized squared error improved stability. Note this meta-objective is *only* used during meta-training and *not* used when applying the learned update rule. The inner loop computation is performed without labels via the UnsupervisedUpdate.

# 4 TRAINING THE UPDATE RULE

## 4.1 APPROXIMATE GRADIENT BASED TRAINING

We choose to meta-optimize via SGD as opposed to reinforcement learning or other black box methods, due to the superior convergence properties of SGD in high dimensions, and the high dimensional nature of $\theta$. Training and computing derivatives through long recurrent computation of this form is notoriously difficult (Pascanu et al., 2013). To improve stability and reduce the computational cost we approximate the gradients $\frac{\partial[\text{MetaObjective}]}{\partial\theta}$ via truncated backprop through time (Shaban et al., 2018). Many additional design choices were also crucial to achieving stability and convergence in meta-learning, including the use of batch norm, and restricting the norm of the UnsupervisedUpdate update step (a full discussion of these and other choices is in Appendix B).

## 4.2 META-TRAINING DISTRIBUTION AND GENERALIZATION

Generalization in our learned optimizer comes from both the form of the UnsupervisedUpdate (Section 3.2), and from the meta-training distribution. Our meta-training distribution is composed of both datasets and base model architectures.

We construct a set of training tasks consisting of CIFAR10 (Krizhevsky and Hinton, 2009) and multi-class classification from subsets of classes from Imagenet (Russakovsky et al., 2015) as well as from a dataset consisting of rendered fonts (Appendix H.1.1). We find that increased training dataset variation actually improves the meta-optimization process. To reduce computation we restrict the input data to 16x16 pixels or less during meta-training, and resize all datasets accordingly. For evaluation, we use MNIST (LeCun et al., 1998), Fashion MNIST (Xiao et al., 2017), IMDB (Maas et al., 2011), and a hold-out set of Imagenet classes. We additionally sample the base model architecture. We sample number of layers uniformly between 2-5 and the number of units per layer logarithmically between 64 to 512.

As part of preprocessing, we permute all inputs along the feature dimension, so that the UnsupervisedUpdate must learn a permutation invariant learning rule. Unlike other work, we focus explicitly on learning a learning algorithm as opposed to the discovery of fixed feature extractors that generalize across similar tasks. This makes the learning task *much* harder, as the UnsupervisedUpdate has to discover the relationship between pixels based solely on their joint statistics, and cannot "cheat" and memorize pixel identity. To provide further dataset variation, we additionally augment the data with shifts, rotations, and noise. We add these augmentation coefficients as additional regression targets for the meta-objective–e.g. rotate the image and predict the rotation angle as well as the image class. For additional details, see Appendix H.1.1.

### 4.3 DISTRIBUTED IMPLEMENTATION

We implement the above models in distributed TensorFlow (Abadi et al., 2016). Training uses 512 workers, each of which performs a sequence of partial unrolls of the inner loop UnsupervisedUpdate, and computes gradients of the meta-objective asynchronously. Training takes ∼8 days, and consists of ∼200 thousand updates to $\theta$ with minibatch size 256. Additional details are in Appendix C.

## 5 EXPERIMENTAL RESULTS

First, we examine limitations of existing unsupervised and meta learning methods. Then, we show meta-training and generalization properties of our learned optimizer and finally we conclude by visualizing how our learned update rule works. For details of the experimental setup, see Appendix H.

### 5.1 OBJECTIVE FUNCTION MISMATCH AND EXISTING META-LEARNING METHODS

To illustrate the negative consequences of objective function mismatch in unsupervised learnin algorithms, we train a variational autoencoder on 16x16 CIFAR10. Over the course of training we evaluate classification performance from few shot classification using the learned latent representations. Training curves can be seen in Figure 2. Despite continuing to improve the VAE objective throughout training (not shown here), the classification accuracy decreases sharply later in training.

To demonstrate the reduced generalization that results from learning transferable features rather than an update algorithm, we train a prototypical network (Snell et al., 2017) with and without the input shuffling described in Section 4.2. As the prototypical network primarily learns transferrable features, performance is significantly hampered by input shuffling. Results are in Figure 2.

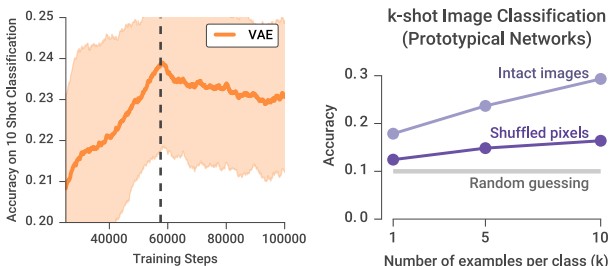

Figure 2: **Left:** Standard unsupervised learning approaches suffer from objective function missmatch. Continuing to optimize a variational auto-encoder (VAE) hurts few-shot accuracy after some number of steps (dashed line). **Right:** Prototypical networks transfer features rather than a learning algorithm, and perform poorly if tasks don't have consistent data structure. Training a prototypical network with a fully connected architecture (same as our base model) on a MiniImagenet 10-way classification task with either intact inputs (light purple) or by permuting the pixels before every training and testing task (dark purple). Performance with permuted inputs is greatly reduced (gray line). Our performance is invariant to pixel permutation.

### 5.2 META-OPTIMIZATION

While training, we monitor a rolling average of the meta-objective averaged across all datasets, model architectures, and the number of unrolling steps performed. In Figure 3 the training loss is continuing to decrease after 200 hours of training, which suggests that the approximate training techniques still produce effective learning. In addition to this global number, we measure performance obtained by rolling out the UnsupervisedUpdate on various meta-training and meta-testing datasets. We see that on held out image datasets, such as MNIST and Fashion Mnist, the evaluation loss is still decreasing. However, for datasets in a different domain, such as IMDB sentiment prediction (Maas et al., 2011), we start to see meta-overfitting. For all remaining experimental results, unless otherwise stated, we use meta-parameters, $\theta$, for the UnsupervisedUpdate resulting from 200 hours of meta-training.

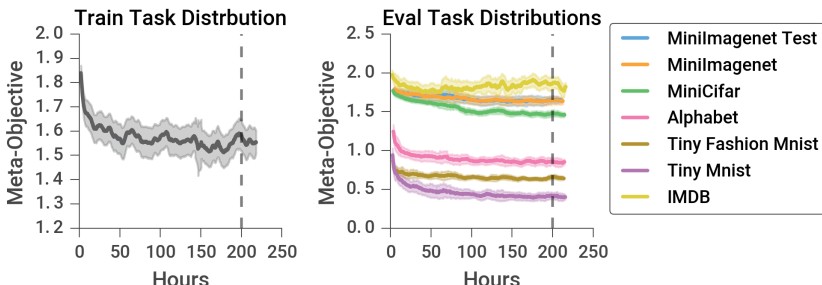

Figure 3: Training curves for the training and evaluation task distributions. Our train set consists of MiniImagenet, Alphabet, and MiniCIFAR. Our test sets are Mini Imagenet Test, Tiny Fashion MNIST, Tiny MNIST and IMDB. Error bars denote standard deviation of evaluations with a fixed window of samples evaluated from a single model. Dashed line at 200 hours indicates model used for remaining experiments unless otherwise stated. For a bigger version of this figure, see Appendix E.

## 5.3 GENERALIZATION

The goal of this work is to learn a general purpose unsupervised representation learning algorithm. As such, this algorithm must be able to generalize across a wide range of scenarios, including tasks that are not sampled i.i.d. from the meta-training distribution. In the following sections, we explore a subset of the factors we seek to generalize over.

**Generalizing over datasets and domains**

In Figure 4, we compare performance on few shot classification with 10 examples per class. We evaluate test performance on holdout datasets of MNIST and Fashion MNIST at 2 resolutions: $14 \times 14$ and $28 \times 28$ (larger than any dataset experienced in meta-training). On the same base model architecture, our learned UnsupervisedUpdate leads to performance better than a variational autoencoder, supervised learning on the labeled examples, and random initialization with trained readout layer.

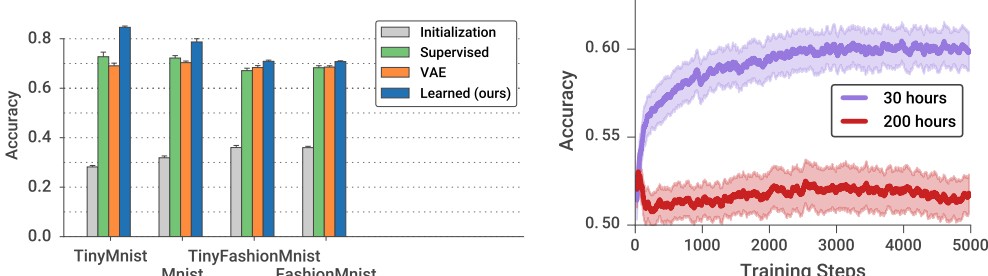

Figure 4: **Left:** The learned UnsupervisedUpdate generalizes to unseen datasets. Our learned update rule produces representations more suitable for few shot classification than those from random initialization or a variational autoecoder and outperforms fully supervised learning on the same labeled examples. Error bars show standard error. **Right:** Early in meta-training (purple), the UnsupervisedUpdate is able to learn useful features on a 2 way text classification data set, IMDB, despite being meta-trained only from image datasets. Later in meta-training (red) performance drops due to the domain mismatch. We show inner-loop training, consisting of 5k applications of the UnsupervisedUpdate evaluating the MetaObjective each iteration. Error bars show standard error across 10 runs.

To further explore generalization limits, we test our learned optimizer on data from a vastly different domain. We train on a binary text classification dataset: IMDB movie reviews (Maas et al., 2011), encoded by computing a bag of words with 1K words. We evaluate using a model 30 hours and 200 hours into meta-training (see Figure 4). Despite being trained exclusively on image datasets, the 30 hour learned optimizer improves upon the random initialization by almost 10%. When meta-training for longer, however, the learned optimizer "meta-overfits" to the image domain resulting in poor performance. This performance is quite low in an absolute sense, for this task. Nevertheless, we find

this result very exciting as we are unaware of any work showing this kind of transfer of learned rules from images to text.

**Generalizing over network architectures**

We train models of varying depths and unit counts with our learned optimizer and compare results at different points in training (Figure 5). We find that despite only training on networks with 2 to 5 layers and 64 to 512 units per layer, the learned rule generalizes to 11 layers and 10,000 units per layer.

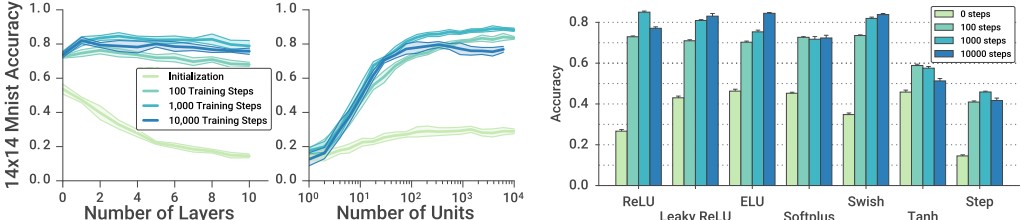

Figure 5: **Left:** The learned UnsupervisedUpdate is capable of optimizing base models with hidden sizes and depths outside the meta-training regime. As we increase the number of units per layer, the learned model can make use of this additional capacity despite never having experienced it during meta-training. **Right:** The learned UnsupervisedUpdate generalizes across many different activation functions not seen in training. We show accuracy over the course of training on 14x14 MNIST.

Next we look at generalization over different activation functions. We apply our learned optimizer on base models with a variety of different activation functions. Performance evaluated at different points in training (Figure 5). Despite training only on ReLU activations, our learned optimizer is able to improve on random initializations in all cases. For certain activations, leaky ReLU (Maas et al., 2013) and Swish (Ramachandran et al., 2017), there is little to no decrease in performance. Another interesting case is the step activation function. These activations are traditionally challenging to train as there is no useful gradient signal. Despite this, our learned UnsupervisedUpdate is capable of optimizing as it does not use base model gradients, and achieves performance double that of random initialization.

## 5.4    How it Learns and How it Learns to Learn

To analyze how our learned optimizer functions, we analyze the first layer filters over the course of meta-training. Despite the permutation invariant nature of our data (enforced by shuffling input image pixels before each unsupervised training run), the base model learns features such as those shown in Figure 6, which appear template-like for MNIST, and local-feature-like for CIFAR10. Early in training, there are coarse features, and a lot of noise. As the meta-training progresses, more interesting and local features emerge.

In an effort to understand what our algorithm learns to do, we fed it data from the two moons dataset. We find that despite being a 2D dataset, dissimilar from the image datasets used in meta-training, the learned model is still capable of manipulating and partially separating the data manifold in a purely unsupervised manner (Figure 6). We also find that almost all the variance in the embedding space is dominated by a few dimensions. As a comparison, we do the same analysis on MNIST. In this setting, the explained variance is spread out over more of the principal components. This makes sense as the generative process contains many more latent dimensions – at least enough to express the 10 digits.

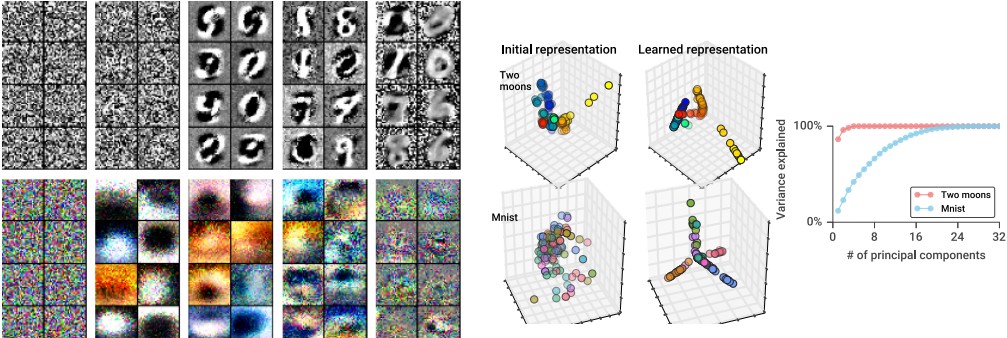

Figure 6: **Left:** From left to right we show first layer base model receptive fields produced by our learned UnsupervisedUpdate rule over the course of meta-training. Each pane consists of first layer filters extracted from $\phi$ after 10k applications of UnsupervisedUpdate on MNIST (top) and CIFAR10 (bottom). For MNIST, the optimizer learns image-template-like features. For CIFAR10, low frequency features evolve into higher frequency and more spatially localized features. For more filters, see Appendix D. **Center:** Visualization of learned representations before (left) and after (right) training a base model with our learned UnsupervisedUpdate for two moons (top) and MNIST (bottom). The UnsupervisedUpdate is capable of manipulating the data manifold, without access to labels, to separate the data classes. Visualization shows a projection of the 32-dimensional representation of the base network onto the top three principal components. **Right:** Cumulative variance explained using principal components analysis (PCA) on the learned representations. The representation for two moons data (red) is much lower dimensional than MNIST (blue), although both occupy a fraction of the full 32-dimensional space.

## 6 DISCUSSION

In this work we meta-learn an unsupervised representation learning update rule. We show performance that matches or exceeds existing unsupervised learning on held out tasks. Additionally, the update rule can train models of varying widths, depths, and activation functions. More broadly, we demonstrate an application of meta-learning for learning complex optimization tasks where no objective is explicitly defined. Analogously to how increased data and compute have powered supervised learning, we believe this work is a proof of principle that the same can be done with algorithm design–replacing hand designed techniques with architectures designed for learning and learned from data via meta-learning.

ACKNOWLEDGMENTS

We would like to thank Samy Bengio, David Dohan, Keren Gu, Gamaleldin Elsayed, C. Daniel Freeman, Sam Greydanus, Nando de Freitas, Ross Goroshin, Ishaan Gulrajani, Eric Jang, Hugo Larochelle, Jeremy Nixon, Esteban Real, Suharsh Sivakumar, Pavel Sountsov, Alex Toshev, George Tucker, Hoang Trieu Trinh, Olga Wichrowska, Lechao Xiao, Zongheng Yang, Jiaqi Zhai and the rest of the Google Brain team for extremely helpful conversations and feedback on this work.

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

## A   More Detailed System Diagram

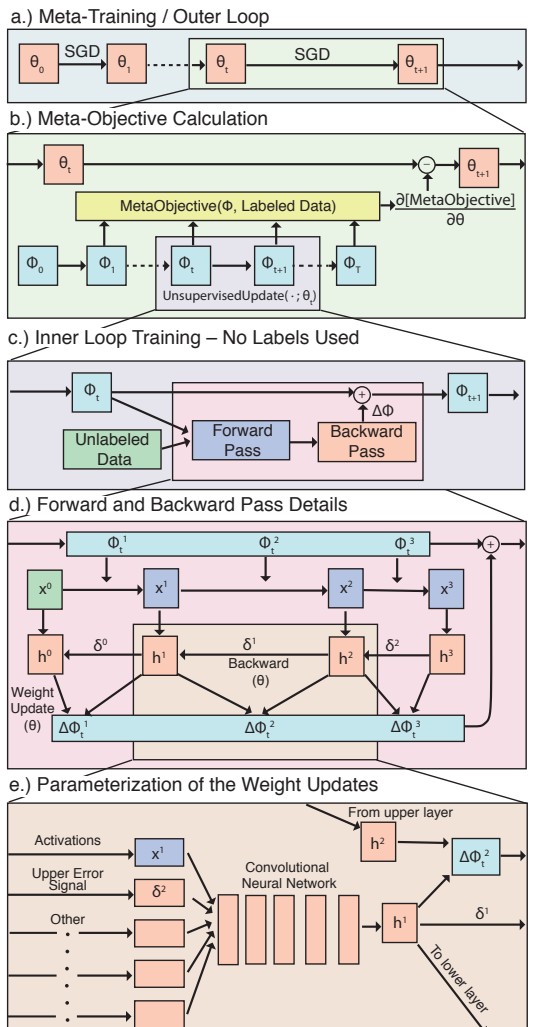

Figure App.1: Schematic for meta-learning an unsupervised learning algorithm. We show the hierarchical nature of both the meta-training procedure and update rule. **a)** Meta-training, where the meta-parameters, $\theta$, are updated via our meta-optimizer (SGD). **b)** The gradients of the MetaObjective with respect to $\theta$ are computed by backpropagation through the unrolled application of the UnsupervisedUpdate. **c)** UnsupervisedUpdate updates the base model parameters ($\phi$) using a minibatch of unlabeled data. **d)** Each application of UnsupervisedUpdate involves computing a forward and "backward" pass through the base model. The base model itself is a fully connected network producing hidden states $x^l$ for each layer $l$. The "backward" pass through the base model uses an error signal from the layer above, $\delta$, which is generated by a meta-learned function. **e.)** The weight updates $\Delta\phi$ are computed using a convolutional network, using $\delta$ and $x$ from the pre- and post-synaptic neurons, along with several other terms discussed in the text.

## B   Stabilizing gradient based meta-learning training

Training and computing derivatives through recurrent computation of this form is notoriously difficult Pascanu et al. (2013). Training parameters of recurrent systems in general can lead to chaos. We used the usual techniques such as gradient clipping (Pascanu et al., 2012), small learning rates, and adaptive learning rate methods (in our case Adam (Kingma and Ba, 2014)), but in practice this

was not enough to train most UnsupervisedUpdate architectures. In this section we address other techniques needed for stable convergence.

When training with truncated backprop the problem shifts from pure optimization to something more like optimizing on a Markov Decision Process where the state space is the base-model weights, $\phi$, and the 'policy' is the learned optimizer. While traversing these states, the policy is constantly meta-optimized and changing, thus changing the distribution of states the optimizer reaches. This type of non-i.i.d training has been discussed at great length with respect to on and off-policy RL training algorithms (Mnih et al., 2013). Other works cast optimizer meta-learning as RL (Li and Malik, 2017) for this very reason, at a large cost in terms of gradient variance. In this work, we partially address this issue by training a large number of workers in parallel, to always maintain a diverse set of states when computing gradients.

For similar reasons, the number of steps per truncation, and the total number of unsupervised training steps, are both sampled in an attempt to limit the bias introduced by truncation.

We found restricting the maximum inner loop step size to be crucial for stability. Pearlmutter (1996) studied the effect of learning rates with respect to the stability of optimization and showed that as the learning rate increases gradients become chaotic. This effect was later demonstrated with respect to neural network training in Maclaurin et al. (2015). If learning rates are not constrained, we found that they rapidly grew and entered this chaotic regime.

Another technique we found useful in addressing these problems is the use of batch norm in both the base model and in the UnsupervisedUpdate rule. Multi-layer perceptron training traditionally requires very precise weight initialization for learning to occur. Poorly scaled initialization can make learning impossible (Schoenholz et al., 2016). When applying a learned optimizer, especially early in meta-training of the learned optimizer, it is very easy for the learned optimizer to cause high variance weights in the base model, after which recovery is difficult. Batch norm helps solve this issues by making more of the weight space usable.

## C  DISTRIBUTED IMPLEMENTATION

We implement the described models in distributed Tensorflow (Abadi et al., 2016). We construct a cluster of 512 workers, each of which computes gradients of the meta-objective asynchronously. Each worker trains on one task by first sampling a dataset, architecture, and a number of training steps. Next, each worker samples $k$ unrolling steps, does $k$ applications of the UnsupervisedUpdate$(\cdot; \theta)$, computes the MetaObjective on each new state, computes $\frac{\partial[\text{MetaObjective}]}{\partial \theta}$ and sends this gradient to a parameter server. The final base-model state, $\phi$, is then used as the starting point for the next unroll until the specified number of steps is reached. These gradients from different workers are batched and $\theta$ is updated with asynchronous SGD. By batching gradients as workers complete unrolls, we eliminate most gradient staleness while retaining the compute efficiency of asynchronous workers, especially given heterogeneous workloads which arise from dataset and model size variation. An overview of our training can be seen in algorithm F. Due to the small base models and the sequential nature of our compute workloads, we use multi core CPUs as opposed to GPUs. Training occurs over the course of ~8 days with ~200 thousand updates to $\theta$ with minibatch size 256.

# D   MORE FILTERS OVER META-TRAINING

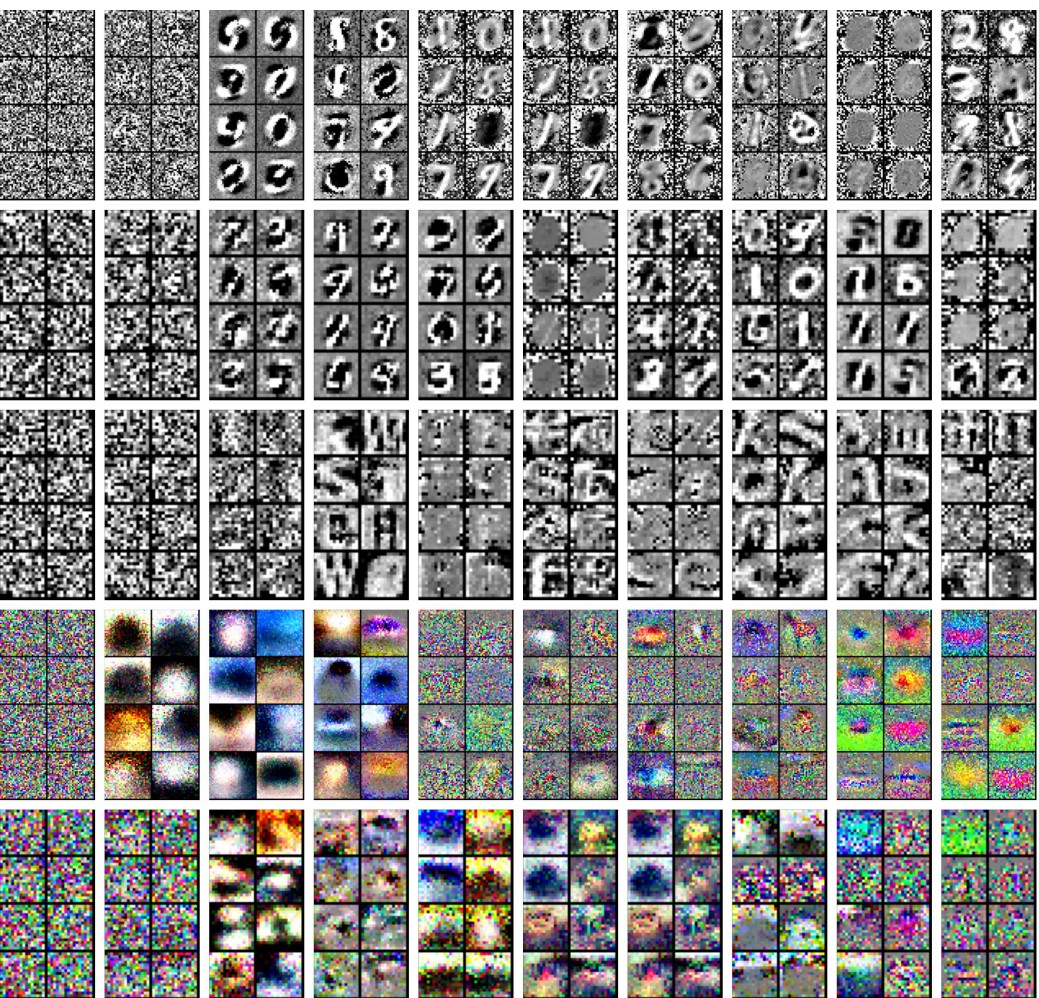

Figure App.2: More filters extracted over the course of meta-training. Note, due to implementation reasons, the columns do not represent the same point / same iteration of $\theta$. Each filter is extracted after 10k inner loop optimization steps, at $\phi_{10k}$. From top to bottom we show: MNIST, Tiny MNIST, Alphabet, CIFAR10, and Mini CIFAR10. Filters shift from noise at initialization to more local features later in meta-training.

# E  META-TRAINING CURVES

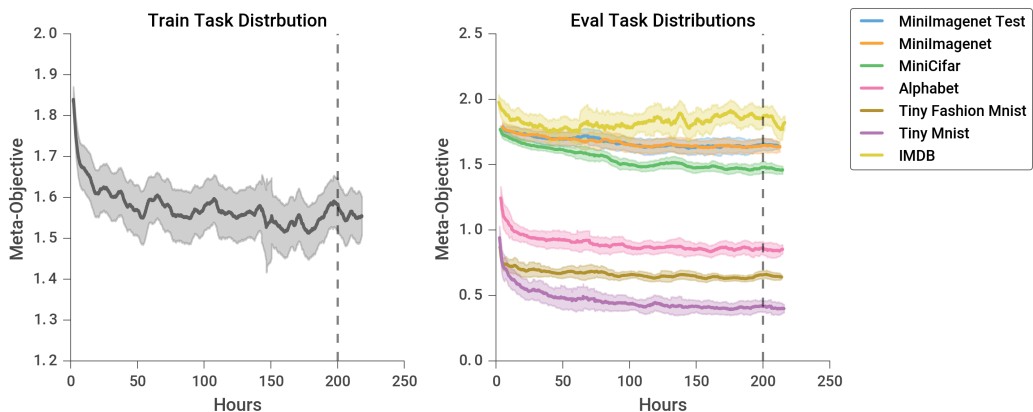

Figure App.3: Training curves for the meta-training and meta-evaluation task distributions. Our meta-train set consists of Mini Imagenet, Alphabet, and MiniCIFAR10. Our meta-test sets are Mini Imagenet Test, Tiny Fashion MNIST, Tiny MNIST and IMDB. Error bars denote standard deviation of evaluations with a fixed window of samples evaluated from a single model.

# F  LEARNING ALGORITHM

Initialize UnsupervisedUpdates parameters, $\theta_0$.
Initialize meta-training step count $v \leftarrow 0$.
Initialize shared gradient state $\mathcal{S}$
Initialize $R$ to be max meta-training steps.
**while** $r < R$ on 512 workers in parallel **do**
    Sample supervised task, $\mathcal{D}$
    Sample base model $f(\cdot; \phi)$ and initialize $\phi_0$ randomly
    Sample $K$ truncated iterations
    Initialize learner iteration count: $t \leftarrow 0$
    **for** $k = 1$ **to** $K$ **do**
      Sample $U$ unroll iterations
      **for** $u = 0$ **to** $U$ **do**
        Sample data, $x$, from $\mathcal{D}$
        $\phi_{t+u+1} = \text{UnsupervisedUpdate}(x, \phi_{t+u}; \theta_r)$
      **end for**
      Initialize MetaObjective accumulator $J \leftarrow 0$
      **for** $u = 1$ **to** $U$ **do**
        **for** $m = 1$ **to** $M$ (number of MetaObjective evaluations per iteration) **do**
          Sample 2 batches of data, $x,x'$, and labels, $y,y'$ from $\mathcal{D}$ for both training and testing.
          $J = J + \frac{1}{UM} \text{MetaObjective}(x, y, x', y', \phi_{t+u})$
        **end for**
      **end for**
      Compute $\frac{\partial J}{\partial \theta_r}$ for the last $U$ steps and store in $\mathcal{S}$
      Update current unroll iteration: $t \leftarrow t + U$
      **if** size of $\mathcal{S}$ > meta-batch-size **then**
        Take averaged batch of gradients $G$ from $\mathcal{S}$
        $\theta_{r+1} = \theta_r - \text{AdamUpdate}(G)$
        $r \leftarrow r + 1$
      **end if**
    **end for**
**end while**

**Algorithm 1:** Distributed Training Algorithm

## G  MODEL SPECIFICATION

In this section we describe the details of our base model and learned optimizer. First, we describe the inner loop, and then we describe the meta-training procedure in more depth.

In the following sections, $\lambda_{name}$ denotes a hyper-parameter for the learned update rule.

The design goals of this system are stated in the text body. At the time of writing, we were unaware of any similar systems, so as such the space of possible architectures was massive. Future work consists of simplifying and improving abstractions around algorithmic components.

An open source implementation of the UnsupervisedUpdate can be found at `https://github.com/tensorflow/models/tree/master/research/learning_unsupervised_learning`.

### G.1  INNER LOOP

The inner loop computation consists of iterative application of the UnsupervisedUpdate on a Base Model parameterized by $\phi$,

$$\phi_{t+1} = \text{UnsupervisedUpdate}(\cdot, \phi_t; \theta), \tag{App.1}$$

where $\phi$ consists of forward weights, $W^l$, biases, $b^l$, parameterizing a multi-layer perceptron as well as backward weights, $V^l$ used by UnsupervisedUpdate when inner-loop training.

This computation can be broken down further as a forward pass on an unlabeled batch of data,

$$x^0 \sim \mathcal{D} \tag{App.2}$$

$$\{x^1..x^L, z^1..z^L\} = f(x^0; \phi_t), \tag{App.3}$$

where $z^l$, and $x^l$ are the pre- and post-activations on layer $l$, and $x^0$ is input data. We then compute weight updates:

$$\{(\Delta W^{1\cdots L})_t, (\Delta b^{1\cdots L})_t, (\Delta V^{1\cdots L})_t\} = \text{ComputeDeltaWeight}(x^0 \cdots x^L, z^1 \cdots z^L, \phi_t; \theta) \tag{App.4}$$

Finally, the next set of $\phi$ forward weights ($W^l$), biases ($b^l$), and backward weights ($V^l$), are computed. We use an SGD like update but with the addition of a decay term. Equivalently, this can be seen as setting the weights and biases to an exponential moving average of the $\Delta W^l$, $\Delta V^l$, and $\Delta b^l$ terms.

$$W_{t+1}^l = W_t^l(1 - \lambda_{\phi lr}) + \Delta W^l \lambda_{\phi lr} \tag{App.5}$$

$$V_{t+1}^l = V_t^l(1 - \lambda_{\phi lr}) + \Delta V^l \lambda_{\phi lr} \tag{App.6}$$

$$b_{t+1}^l = b_t^l(1 - \lambda_{\phi lr}) + \Delta b^l \lambda_{\phi lr} \tag{App.7}$$

We use $\lambda_{\phi lr} = 3e - 4$ in our work. $\Delta W^l$, $\Delta V^l$, $\Delta b^l$ are computed via meta-learned functions (parameterized by $\theta$).

In the following sections, we describe the functional form of the base model, $f$, as well as the functional form of ComputeDeltaWeight $(\cdot; \theta)$.

### G.2  BASE MODEL

The base model, the model our learned update rule is training, is an $L$ layer multi layer perception with batch norm. We define $\phi$ as this model's parameters, consisting of weights ($W^l$) and biases ($b^l$) as well as the backward weights ($V^l$) used only during inner-loop training (applications of UnsupervisedUpdate). We define $N^1..N^L$ to be the sizes of the layers of the base model and $N^0$ to be the size of the input data.

$$\phi = \{W^1..W^L, V^1..V^L, b^1..b^L\}, \tag{App.8}$$

where

$$W^l \in \mathbb{R}^{N^{l-1}, N^l} \tag{App.9}$$

$$V^l \in \mathbb{R}^{N^{l-1}, N^l} \tag{App.10}$$

$$b^l \in \mathbb{R}^{N^l} \tag{App.11}$$

where $N^l$ is the hidden size of the network, $N^0$ is the input size of data, and $N^L$ is the size of the output embedding. In this work, we fix the output layer: $N^L = 32$ and vary the remaining the intermediate $N^{1..(L-1)}$ hidden sizes.

The forward computation parameterized by $\phi$, consumes batches of unlabeled data from a dataset $\mathcal{D}$:

$$x^0 \sim \mathcal{D}, x^0 \in \mathbb{R}^{B, N^0} \tag{App.12}$$

$$z^l = \text{BatchNorm}\left(x^{l-1}W^l\right) + b^l \tag{App.13}$$

$$x^l = \text{ReLU}(z^l) \tag{App.14}$$

for $l = 1..L$.

We define $f(x; \phi)$ as a function that returns the set of internal pre- and post-activation function hidden states as well as $\hat{f}(x; \phi)$ as the function returning the final hidden state:

$$f(x, \phi) = \{z^1...z^L, x^0..x^L, \phi\} \tag{App.15}$$

$$\hat{f}(x, \phi) = x^L \tag{App.16}$$

### G.3 METAOBJECTIVE

We define the MetaObjective to be a few shot linear regression. To increase stability and avoid undesirable loss landscapes, we additionally center, as well as normalize the predicted target before doing the loss computation. The full computation is as follows:

$$\text{MetaObjective}(x, y, x', y', \phi) :: (\mathbb{R}^{B, N^0}, \mathbb{R}^{B, N^{classes}}, \mathbb{R}^{B, N^0}, \mathbb{R}^{B, N^{classes}}, \Phi) \to \mathbb{R}^1, \tag{App.17}$$

where $N^{classes}$ is the number of classes, and $y, y'$ are one hot encoded labels.

First, the inputs are converted to embeddings with the base model,

$$x^L = \hat{f}(x; \phi) \tag{App.18}$$

$$x'^L = \hat{f}(x'; \phi) \tag{App.19}$$

Next, we center and normalize the prediction targets. We show this for $y$, but $y'$ is processed identically.

$$\bar{y} = \frac{1}{BN^{classes}} \sum_i^B \sum_j^{N^{classes}} y_{ij} \tag{App.20}$$

$$\hat{y}_{ij} = \frac{y_{ij} - \bar{y}}{\sqrt{\frac{1}{N^{classes}} \sum_a^{N^{classes}} \|\bar{y}_{ia}\|_2^2}} \tag{App.21}$$

We then solve for the linear regression weights in closed form with features: $x^L$ and targets: $\hat{y}$. We account for a bias by concatenating a 1's vector to the features.

$$A = [x^L; \mathbb{1}] \tag{App.22}$$

$$C = \left((A^t A) + I\lambda_{\text{ridge}}\right)^{-1} A^T \hat{y} \tag{App.23}$$

We then use these inferred regression weights $C$ to make a prediction on the second batch of data, normalize the resulting prediction, and compute a final loss,

$$p = C[x'^L; \mathbb{1}] \tag{App.24}$$

$$\hat{p}_{bi} = \frac{p_{bi}}{\|p_b\|_2} \tag{App.25}$$

$$\text{MetaObjective}(\cdot) = \frac{1}{B} \sum_b^B \|\hat{p}_b - \hat{y}_b\|_2^2. \tag{App.26}$$

Note that due to the normalization of predictions and targets, this corresponds to a cosine distance (up to an offset and multiplicative factor).

## G.4 UNSUPERVISEDUPDATE

The learned update rule is parameterized by $\theta$. In the following section, we denote all instances of $\theta$ with a subscript to be separate named learnable parameters of the UnsupervisedUpdate. $\theta$ is shared across all instantiations of the unsupervised learning rule (shared across layers). It is this weight sharing that lets us generalize across different network architectures.

The computation is split into a few main components: first, there is the forward pass, defined in $f(x; \phi)$. Next, there is a "backward" pass, which operates on the hidden states in a reverse order to propagate an error signal $\delta^l$ back down the network. In this process, a hidden state $h^l$ is created for each layer. These h are tensors with a batch, neuron and feature index: $h^l \in \mathbb{R}^{B, N^l, \lambda_{hdims}}$ where $\lambda_{hdims} = 64$. Weight updates to $W^l$ and $b^l$ are then readout from these $h^l$ and other signals found both locally, and in aggregate along both batch and neuron.

### G.4.1 BACKWARD ERROR PROPAGATION

In backprop, there exists a single scalar error signal that gets backpropagated back down the network. In our work, this error signal does not exist as there is no loss being optimized. Instead we have a learned top-down signal, $d^L$, at the top of the network. Because we are no longer restricted to the form of backprop, we make this quantity a vector for each unit in the network, rather than a scalar,

$$x^0 \sim \mathcal{D}, x^0 \in \mathbb{R}^{B, N_x} \tag{App.27}$$

$$\{z^1...z^L, x^0..x^L, \phi\} = f(x; \phi) \tag{App.28}$$

$$d^L = \text{TopD}(x^L; \theta_{topD}) \tag{App.29}$$

$$\tag{App.30}$$

where $d^l \in \mathbb{R}^{B, N^l, \lambda_{deltadims}}$. In this work we set $\lambda_{deltadims} = 32$. The architecture of $\text{TopD}$ is a neural network that operates along every dimension of $x^L$. The specification can be found in G.7.

We structure our error propagation similarly to the structure of the backpropagated error signal in a standard MLP with contributions to the error at every layer in the network,

$$\delta^l_{ijd} = d^l_{ijd} \odot \sigma(z^l_{ijd}) + \sum_k^{\lambda_{hdims}} (\theta_{errorPropW})_{kd} h^l_{ijk} + (\theta_{errorPropB})_d \tag{App.31}$$

where $\delta^l$ has the same shape as $d^l$, and $\theta_{errorPropW} \in \mathbb{R}^{\lambda_{hdims}, \lambda_{deltadims}}$ and $\theta_{errorPropB} \in \mathbb{R}^{\lambda_{deltadims}}$. Note both $\theta_{errorPropW}$ and $\theta_{errorPropB}$ are shared for all layers $l$.

In a similar way to backprop, we move this signal down the network via multiplying by a backward weight matrix ($V^l$). We do not use the previous weight matrix transpose as done in backprop, instead we learn a separate set of weights that are not tied to the forward weights and updated along with the forward weights as described in G.5. Additionally, we normalize the signal to have fixed second moment,

$$\tilde{d}^l_{imd} = \sum_j^{N^{l+1}} \delta^{l+1}_{ijd} (V^{l+1})_{mj} \tag{App.32}$$

$$d^l_{imd} = \hat{d}^l_{imd} \left( \frac{1}{\lambda_{deltadims}} \sum_a^{\lambda_{deltadims}} \tilde{d}^l_{ima} \right)^{-\frac{1}{2}} \tag{App.33}$$

The internal $h^l \in \mathbb{R}^{B, N^l, \lambda_{hdims}}$ vectors are computed via:

$$h^l = \text{ComputeH}\left(d^l, x^l, z^l; \theta_{computeH}\right) \tag{App.34}$$

The architecture of $\text{ComputeH}$ is a neural network that operates on every dimension of all the inputs. It can be found in G.8. These definitions are recursive, and are computed in order: $h^L, h^{L-1} \cdots h^1, h^0$. With these computed, weight updates can be read out (Section G.5). When the corresponding symbols are not defined (e.g. $z^0$) a zeros tensor with the correct shape is used instead.

## G.5 WEIGHT UPDATES

The following is the implementation of $\text{ComputeDeltaWeight}(x^0 \cdots x^L, z^1 \cdots z^L, \phi; \theta)$

For a given layer, $l$, our weight updates are a mixture of multiple low rank readouts from $h^l$ and $h^{l-1}$. These terms are then added together with a learnable weight, in $\theta$ to form a final update. The final update is then normalized mixed with the previous weights. We update both the forward, $W^l$, and the backward, $V^l$, using the same update rule parameters $\theta$. We show the forward weight update rule here, and drop the backward for brevity.

For convenience, we define a low rank readout function $\text{LowRR}$ that takes $h^l$ like tensors, and outputs a single lower rank tensor.

$$\text{LowRR}\left(h^a, h^b; \Theta\right) :: \tag{App.35}$$

$$\left(\mathbb{R}^{B, N^a, \lambda_{hdims}}, \mathbb{R}^{B, N^b, \lambda_{hdims}}\right) \to \mathbb{R}^{N''^a, N^b} \tag{App.36}$$

Here, $\Theta$, is a placehoder for the parameters of the given readout. $\text{LowRR}$ is defined as:

$$\Theta = \{P^a, P^b\} \quad \text{where} \quad P^a \in \mathbb{R}^{\lambda_{hdims}, \lambda_{gradc}} \quad \text{and} \quad P^b \in \mathbb{R}^{\lambda_{hdims}, \lambda_{gradc}} \tag{App.37}$$

$$r^a_{ijp} = \sum_k h^a_{ijk} P^a_{kp} \tag{App.38}$$

$$r^b_{ijp} = \sum_k h^b_{ijk} P^b_{kp} \tag{App.39}$$

$$\text{LowRR}\left(\cdot\right)_{jp} = \sum_i^B \sum_k^{\lambda_{gradc}} r^b_{ijk} r^a_{ipk} \frac{1}{B\lambda_{hdims}} \tag{App.40}$$

where $\lambda_{gradc} = 4$ and is the rank of the readout matrix (per batch element).

### G.5.1 LOCAL TERMS

This sequence of terms allow the weight to be adjusted as a function of state in the pre- and post-synaptic neurons. They should be viewed as a basis function representation of the way in which the weight changes as a function of pre- and post-synaptic neurons, and the current weight value. We express each of these contributions to the weight update as a sequence of weight update planes, with the $i$th plane written $\Delta W^l_i \in \mathbb{R}^{N^{l-1} \times N^l}$. Each of these planes will be linearly summed, with coefficients generated as described in Equation App.57, in order to generate the eventual weight update.

$$\hat{W}^l = \frac{W^l}{\sqrt{\frac{1}{N^{l-1}} \sum_i^{N^{l-1}} (W^l)^2_i}} \tag{App.41}$$

$$\Delta W^l_1 = \hat{W}^l \tag{App.42}$$

$$\Delta W^l_2 = (\hat{W}^l)^2 sign(\hat{W}^l) \tag{App.43}$$

$$\Delta W^l_3 = \text{LowRR}(h^{l-1}, h^l; \theta_{zero}) \tag{App.44}$$

$$\Delta W^l_4 = \exp(-(\hat{W}^l_b)^2) \odot \text{LowRR}(h^{l-1}, h^l; \theta_{rbf}) \tag{App.45}$$

$$\Delta W^l_5 = W^l_b \odot \text{LowRR}(h^{l-1}, h^l; \theta_{first}) \tag{App.46}$$

$$\Delta W^l_6 = \frac{1}{B} \sum_b^B \left(x^{l-1}_b - \frac{1}{B} \sum_{b'}^B x^{l-1}_{b'}\right)^T \left(x^l_b - \frac{1}{B} \sum_{b'}^B x^l_{b'}\right)^T \tag{App.47}$$

### G.5.2 DECORRELATION TERMS

Additional weight update planes are designed to aid units in remaining decorrelated from each other's activity, and in decorrelating their receptive fields. Without terms like this, a common failure mode

is for many units in a layer to develop near-identical representations. Here, $S_i^l$ indicates a scratch matrix associated with weight update plane $i$ and layer $l$.

$$S_7^l = \frac{1}{\sqrt{N^{l-1}}} \text{LowRR}(h^{l-1}, h^{l-1}, \theta_{linLowerSymm}) \tag{App.48}$$

$$\Delta W_7^l = \frac{1}{\sqrt{2}} \left[ \left( S_7^l + \left( S_7^l \right)^T \right) \odot (1 - I) \right] W^l \tag{App.49}$$

$$S_8^l = \frac{1}{\sqrt{N^{l-1}}} \text{LowRR}(h^{l-1}, h^{l-1}, \theta_{sqrLowerSymm}) \tag{App.50}$$

$$\Delta W_8^l = \frac{1}{\sqrt{2}} \left[ \left( S_8^l + \left( S_8^l \right)^T \right) \odot (1 - I) \right] \left( \sqrt{1 + (W^l)^2} - 1 \right) \tag{App.51}$$

$$S_9^l = \frac{1}{\sqrt{N^l}} \text{LowRR}(h^l, h^l, \theta_{linUpperSymm}) \tag{App.52}$$

$$\Delta W_9^l = \frac{1}{\sqrt{2}} W^l \left[ \left( S_9^l + \left( S_9^l \right)^T \right) \odot (1 - I) \right] \tag{App.53}$$

$$S_{10}^l = \frac{1}{\sqrt{N^l}} \text{LowRR}(h^l, h^l, \theta_{sqrUpperSymm}) \tag{App.54}$$

$$\Delta W_{10}^l = \frac{1}{\sqrt{2}} \left( \sqrt{1 + (W^l)^2} - 1 \right) \left[ \left( S_{10}^l + \left( S_{10}^l \right)^T \right) \odot (1 - I) \right] \tag{App.55}$$

$$\tag{App.56}$$

### G.6 Application of the weight terms in the optimizer

We then normalize, re-weight, and merge each of these weight update planes into a single term, which will be used in the weight update step,

$$\Delta \tilde{W}_i^l = \frac{\Delta W_i^l}{\sqrt{1 + \frac{1}{N^{l-1}N^l} \sum_m^{N^{l-1}} \sum_n^{N^l} \left( \Delta W_{imn}^l \right)^2}} \tag{App.57}$$

$$(\Delta W_{merge}^l)_{jk} = \frac{1}{B} \sum_i^B (\theta_{mergeW})_i \Delta \tilde{W}_{ijk}^l, \tag{App.58}$$

where $\theta_{mergeW} \in \mathbb{R}^{10}$ (as we have 10 input planes).

To prevent pathologies during training, we perform two post processing steps to prevent the learned optimizer from cheating, and increasing its effective learning rate, leading to instability. We only allow updates which do not decrease the weight matrix magnitude,

$$\Delta W_{orth}^l = \Delta W_{merge}^l - \hat{W}^l \text{ReLU}(\Delta W_{merge} \cdot \hat{W}^l), \tag{App.59}$$

where $\hat{W}^l$ is $W^l$ scaled to have unit norm, and we normalize the length of the update,

$$\Delta W_{final}^l = \frac{\Delta W_{orth}^l}{\sqrt{1 + \frac{1}{N^{l-1}N^l} \sum_m^{N^{l-1}} \sum_n^{N^l} \left( \Delta W_{orth}^l \right)_{mn}^2}} \tag{App.60}$$

To compute changes in the biases, we do a readout from $h^l$. We put some constraints on this update to prevent the biases from pushing all units into the linear regime, and minimizing learning. We found this to be a possible pathology.

$$\Delta b_{base}^l = \frac{1}{B} \sum_i^B \sum_k^{\lambda_{hdims}} (\theta_{Breadout})_k h_{ijk}^l \tag{App.61}$$

$$\Delta b_{constrained}^l = \Delta b_{base}^l - \text{ReLU} \left( -\frac{1}{N^l} \sum_i^{N^l} \left( \Delta b_{base}^l \right)_i \right), \tag{App.62}$$

where $\theta_{Breadout} \in \mathbb{R}^{\lambda_{hdims}}$.

We then normalize the update via the second moment:

$$\Delta b^l_{final} = \frac{\Delta b^l_{constrained}}{\frac{1}{N^l} \sum_i^{N^l} (\Delta b^l_{constrained})_i^2} \tag{App.63}$$

Finally, we define ComputeDeltaWeight as all layer's forward weight updates, and backward weight updates, and bias updates.

$$\text{ComputeDeltaWeight}(x^0 \cdots x^L, z^1 \cdots z^L, \phi_t; \theta) = (\Delta W^{1..L}_{final}, \Delta b^{1..L}_{final}, \Delta V^{1..L}_{final}) \tag{App.64}$$

## G.7 TopD

This function performs various convolutions over the batch dimension and data dimension. For ease of notation, we use $m$ as an intermediate variable. Additionally, we drop all convolution and batch norm parameterizations. They are all separate elements of $\theta_{topD}$. We define two 1D convolution operators that act on rank 3 tensors: ConvBatch which performs convolutions over the zeroth index, and ConvUnit which performs convolutions along the firs index. We define the $S$ argument to be size of hidden units, and the $K$ argument to be the kernel size of the 1D convolutions. Additionally, we set the second argument of BatchNorm to be the axis normalized over.

$$\text{TopD}\left(x^L; \theta_{topD}\right) :: \mathbb{R}^{B,N^L} \to \mathbb{R}^{B,N^L,\lambda_{deltadims}} \tag{App.65}$$

First, we reshape to add another dimension to the end of $x^L$, in pseudocode:

$$m_0 = [x^L] \tag{App.66}$$

Next, a convolution is performed on the batch dimension with a batch norm and a ReLU non linearity. This starts to pull information from around the batch into these channels.

$$m_1 = \text{ConvBatch}\left(m_0, S = \lambda_{topdeltasize}, K = 5\right) \tag{App.67}$$
$$m_2 = \text{ReLU}\left(\text{BatchNorm}\left(m_1, [0,1]\right)\right) \tag{App.68}$$
$$\tag{App.69}$$

We set $\lambda_{topdeltasize} = 64$. Next, a series of unit convolutions (convolutions over the last dimension) are performed. These act as compute over information composed from the nearby elements of the batch. These unit dimensions effectively rewrite the batch dimension. This restricts the model to operate on a fixed size batch.

$$m_3 = \text{ConvUnit}\left(m_2, S = B, K = 3\right) \tag{App.70}$$
$$m_4 = \text{ReLU}\left(\text{BatchNorm}\left(m_3, [0,1]\right)\right) \tag{App.71}$$
$$m_5 = \text{ConvUnit}\left(m_4, S = B, K = 3\right) \tag{App.72}$$
$$m_6 = \text{ReLU}\left(\text{BatchNorm}\left(m_5, [0,1]\right)\right) \tag{App.73}$$

Next a series of 1D convolutions are performed over the batch dimension for more compute capacity.

$$m_7 = \text{ConvBatch}\left(m_6, S = \lambda_{topdeltasize}, K = 3\right) \tag{App.74}$$
$$m_8 = \text{ReLU}\left(\text{BatchNorm}\left(m_7, [0,1]\right)\right) \tag{App.75}$$
$$m_9 = \text{ConvBatch}\left(m_8, S = \lambda_{topdeltasize}, K = 3\right) \tag{App.76}$$
$$m_{10} = \text{ReLU}\left(\text{BatchNorm}\left(m_9, [0,1]\right)\right) \tag{App.77}$$

Finally, we convert the representations to the desired dimensions and output.

$$m_{11} = \text{ConvBatch}\left(m_{10}, S = \lambda_{deltadims}, K = 3\right) \tag{App.78}$$
$$\text{TopD}\left(x^L; \theta_{topD}\right) = m_{11} \tag{App.79}$$

## G.8    ComputeH

This is the main computation performed while transfering signals down the network and the output is directly used for weight updates. It is defined as:

$$\text{ComputeH}\left(d^l, x^l, z^l, W^l, W^{l+1}, b^l; \theta_{computeH}\right) ::$$
$$\left(\mathbb{R}^{B,N^l,\lambda_{deltadims}}, \mathbb{R}^{B,N^l}, \mathbb{R}^{B,N^l}, \mathbb{R}^{N^{l-1},N^l}, \mathbb{R}^{N^1,N^{l+1}}, \mathbb{R}^{N^l}\right) \rightarrow \left(\mathbb{R}^{B,N^l,\lambda_{hdims}}\right) \quad \text{(App.80)}$$

The outputs of the base model, $(x^l, z^l)$, plus an additional positional embeddings are stacked then concatenated with $d$, to form a tensor in $\mathbb{R}^{B,N^l,(4+\lambda_{deltadims})}$:

$$(p^0)_{ij} = \sin(\frac{2j\pi}{N^l}) \quad \text{(App.81)}$$

$$(p^1)_{ij} = \cos(\frac{2j\pi}{N^l}) \quad \text{(App.82)}$$

$$m_0 = [x^l, z^l, p^0, p^1] \quad \text{where} \quad m_0 \in \mathbb{R}^{B,N^l,4} \quad \text{(App.83)}$$

$$m_1 = [m_0; d^l] \quad \text{(App.84)}$$

Statistics across the batch and unit dimensions, 0 and 1, are computed. We define a $\text{Stats}_i$ function bellow. We have 2 instances for both the zeroth and the first index, shown bellow is the zeroth index and the first is omitted.

$$\text{Stats}_0(w) :: \mathbb{R}^{K^0,K^1} \rightarrow \mathbb{R}^{K^1,4} \quad \text{(App.85)}$$

$$(s_{l1})_j = \frac{1}{K^0} \sum_i^{K^0} \text{abs}(w_{ij}) \quad \text{(App.86)}$$

$$(s_{l2})_j = \sqrt{\frac{1}{K^0} \sum_i^{K^0} (w_{ij})^2} \quad \text{(App.87)}$$

$$(s_\mu)_j = \frac{1}{K^0} \sum_i^{K^0} w_{ij} \quad \text{(App.88)}$$

$$(s_\sigma)_j = \sqrt{\frac{1}{K^0} \sum_i^{K^0} ((s_\mu)_i - w_{ij})^2} \quad \text{(App.89)}$$

$$Stats_0(w) = [s_{l1}, s_{l2}, s_\mu, s_\sigma] \quad \text{(App.90)}$$

We the compute statistics of the weight matrix below, and above. We tile the statistics to the appropriate dimensions and concatenate with normalized inputs as well as with the bias (also tiled appropriately).

$$(s_0)_{ijk} = (\text{Stats}_0(W^l, 0))_j \quad \text{(App.91)}$$

$$(s_1)_{ijk} = (\text{Stats}_1(W^{l+1}, 1))_i \quad \text{(App.92)}$$

$$m_2 = \text{BatchNorm}(m_1, [0,1]) \quad \text{(App.93)}$$

$$\hat{b}_{ijk} = b_j^l \quad \text{where} \quad \hat{b} \in \mathbb{R}^{B,N^l,1} \quad \text{(App.94)}$$

$$m_3 = [s_0; s_1; m_2; \hat{b}] \quad \text{where} \quad m_3 \in \mathbb{R}^{B,N^l,4+4+(4+\lambda_{deltadims})+1} \quad \text{(App.95)}$$

With the inputs prepared, we next perform a series of convolutions on batch and unit dimensions, (0, 1).

$$m_4 = \text{ConvBatch}\left(m_3, S = \lambda_{computehsize}, K = 3\right) \tag{App.96}$$

$$m_5 = \text{ReLU}\left(\text{BatchNorm}\left(m_4, [0, 1]\right)\right) \tag{App.97}$$

$$m_6 = \text{ConvUnit}\left(m_5, S = \lambda_{computehsize}, K = 3\right) \tag{App.98}$$

$$m_7 = \text{ReLU}\left(\text{BatchNorm}\left(m_6, [0, 1]\right)\right) \tag{App.99}$$

$$m_8 = \text{ConvBatch}\left(m_7, S = \lambda_{computehsize}, K = 3\right) \tag{App.100}$$

$$m_9 = \text{ReLU}\left(\text{BatchNorm}\left(m_8, [0, 1]\right)\right) \tag{App.101}$$

$$m_{10} = \text{ConvUnit}\left(m_9, S = \lambda_{computehsize}, K = 3\right) \tag{App.102}$$

$$m_{11} = \text{ReLU}\left(\text{BatchNorm}\left(m_{10}, [0, 1]\right)\right) \tag{App.103}$$

The result is then output.

$$\text{ComputeH}\left(\cdot; \theta_{computeH}\right) = m_{11} \tag{App.104}$$

We set $\lambda_{computehsize} = 64$ which is the inner computation size.

## H  EXPERIMENTAL DETAILS

### H.1  META TRAINING

#### H.1.1  TRAINING DATA DISTRIBUTION

We trained on a data distribution consisting of tasks sampled uniformly over the following datasets. Half of our training tasks where constructed off of a dataset consisting of 1000 font rendered characters. We resized these to 14x14 black and white images. We call this the glyph dataset. "Alphabet" is an example of such a dataset consisting of alphabet characters. We used a mixture 10, 13, 14, 17, 20, and 30 way classification problems randomly sampled, as well as sampling from three 10-way classification problems sampled from specific types of images: letters of the alphabet, math symbols, and currency symbols. For half of the random sampling and all of the specialized selection we apply additional augmentation. This augmentation consists of random rotations (up to 360 degrees) and shifts up to +-5 pixels in the x and y directions. The parameters of the augmentation were inserted into the regression target of the MetaObjective as a curriculum of sorts and to provide diverse training signal.

In addition to the the glyph set, we additionally used Cifar10, resized to 16x16, as well as 10, 15, 20, and 25 way classification problems from imagenet. Once again we resized to 16x16 for compute reasons.

With a dataset selected, we apply additional augmentation with some probability consisting of the following augmentations. A per task dropout mask (fixed mask across all images in that task). A per example dropout mask (a random mask per image). A permutation sampled from a fixed number of pre created permutations per class. A per image random shift in the x direction each of image. All of these additional augmentations help with larger domain transfer.

### H.2  META-OPTIMIZATION

We employ Adam (Kingma and Ba, 2014) as our meta-optimizer. We use a learning rate schedule of 3e-4 for the first 100k steps, then 1e-4 for next 50k steps, then 2e-5 for remainder of meta-training. We use gradient clipping of norm 5 on minibatchs of size 256.

We compute our meta-objective by averaging 5 evaluation of the linear regression. We use a ridge penalty of 0.1 for all this work.

When computing truncated gradients, we initially sample the number of unrolled applications of the UnsupervisedUpdate in a uniform distribution of [2,4]. This is the number of steps gradients are backpropogated through. Over the course of 50k meta-training steps we uniformly increase this to [8,15]. This increases meta-training speed and stability as large unrolls early in training can be unstable and don't seem to provide any value.

For sampling the number of truncated steps (number of times the above unrolls are performed), we use a shifted normal distribution – a normal distribution with the same mean and standard deviation. We chose this based on the expected distribution of the training step, $\phi$ iteration number, across the cluster of workers. We initially set the standard deviation low, 20, but slowly increased it over the course of 5000 steps to 20k steps. This slow increase also improved stability and training speed.

## H.3 EXPERIMENTAL SETUP

For each experimental figure, we document the details.

### H.3.1 OBJECTIVE FUNCTION MISMATCH

The VAE we used consists of 3 layers, size 128, with ReLU activations and batch norm between each layer. We then learn a projection to mean and log std of size 32. We sample, and use the inverse architecture to decode back to images. We use a quantized normal distribution (once again parameterized as mean and log std) as a posterior. We train with Adam with a learning rate of 1e-4. To isolate the effects of objective function mismatch and overfitting, we both train on the unlabeled training set and evaluate on the labeled training set instead of a validation set.

### H.3.2 GENERALIZATION: DATASET AND DOMAIN

We use a 4 layer, size 128 unit architecture with a 32 layer embedding for all models. We select performance at 100k training steps for the VAE, and 3k for our learned optimizer.

Our supervised learning baseline consists of the same architecture for the base model but with an additional layer that outputs log probabilities. We train with cross entropy loss on a dataset consisting of only 10 examples per class (to match the other numbers). Surprisingly, the noise from batch norm acts as a regularizer, and allowing us to avoid needing a complex early stopping scheme as test set performance simply plateaus over the course of 10k steps. We train with Adam with a learning rate of 3e-3, selected via a grid over learning rate on test set performance. In this setting, having a true validation set would dramatically lower the amount of labeled data available (only 100 labeled examples) and using the test set only aids in the this baseline's performance.

For the IMDB experiments, we tokenized and selected the top 1K words with an additional set of tokens for unknown, sentence start, and sentence end. Our encoding consisted of a 1 if the word is present, otherwise 0. We used the same 4 layer, 128 hidden unit MLP with an addition layer outputting a 32 dimensional embedding.

### H.3.3 GENERALIZATION: NETWORK ARCHITECTURE

We used ReLU activations and 4 layers of size 128 with an additional layer to 32 units unless otherwise specified by the specific experiment.

### H.3.4 EXISTING META LEARNING MODELS

We trained prototypical networks (Snell et al., 2017) on either intact or shuffled mini-Imagenet images. For shuffled images, we generated a fixed random permutation for the inputs independently for every instantiation of the base network (or 'episode' in the meta-learning literature (Vinyals et al., 2016; Ravi and Larochelle, 2016)). The purpose of shuffling was to demonstrate the inductive bias of this type of meta-learning, namely that they do not generalize across data domain. Note that the base network trained was the same fully connected architecture like that used in this paper (3 layers, size 128, with ReLU activations and batch normalization between layers). Though the original paper used a convolutional architecture, here we swapped it with the fully connected architecture because the tied weights in a convolutional model do not make sense with shuffled pixels.

