# OpenReview forum: "Meta-Learning Update Rules for Unsupervised Representation Learning"
_ICLR.cc/2019/Conference_

### Official Review · AnonReviewer2 · 2018-11-01
**substantial step towards good unsupervised and local learning**

**Rating:** 8
**Confidence:** 3

**Review:**

The paper describes unsupervised learning as a meta-learning problem: the observation is that unsupervised learning rules are effectively supervised by the quality of the representations that they yield relative to subsequent later semi-supervised (or RL) learning. The learning-to-learning algorithm allows for learning network architecture parameters, and also 'network-in-networks' that determine the unsupervised learning signal based on pre and post activations.

Quality
The proposed algorithm is well defined, and it is compared against relevant competing algorithms on relevant problems.
The results show that the algorithm is competitive with other approaches like VAE (very slightly outperforms).

Clarity
The paper is well written and clearly structured. The section 5.4 is a bit hard to understand, with very very small images.

Originality
There is an extensive literature on meta-learning, which is expanded upon in Appendix A. The main innovation in this work is the parametric update rule for outer loop updates, which does have some similarity to the old work by Bengio in 1990 and 1992.

Significance
- pros clear and seemingly state-of-the-art results, intuitive approach,
-cons only very modestly better than other methods. I would like to get a feel for why VAE is so good tbh (though the authors show that VAE has a problem with objective function mismatch).

One comment: the update rule takes as inputs pre and post activity and a backpropagated error; it seems natural to also use the local gradient of the neuron's transfer function here, as many three or four factor learning rules do.

---

> ### Author Response · Authors · 2018-11-14
> **Thank you**
>
> Thank you for your thoughtful review! Comments below:
>
> "The section 5.4 is a bit hard to understand, with very very small images."
> We apologize for the lack of clarity. We will improve this section and will increase the image size!
>
> "cons only very modestly better than other methods. I would like to get a feel for why VAE is so good tbh (though the authors show that VAE has a problem with objective function mismatch)."
> In generative modeling, understanding what design principles lead to reusable representations is a huge open field of study, but many people have promoted compositional generative models[1,2] and information theoretic measures of how well the model captures structure in the data [3,4]. VAEs possess both of these attributes.
>
> "One comment: the update rule takes as inputs pre and post activity and a backpropagated error; it seems natural to also use the local gradient of the neuron's transfer function here, as many three or four factor learning rules do."
> This is a great suggestion! Thanks.
>
> [1]Yoshua Bengio, Aaron Courville, and Pascal Vincent. Representation learning: A review and new perspectives. 2013.
> [2] Kingma, Diederik P., and Max Welling. "Auto-encoding variational bayes." arXiv preprint arXiv:1312.6114 (2013).
> [2]Hinton, Geoffrey E., et al. "The" wake-sleep" algorithm for unsupervised neural networks." Science 268.5214 (1995): 1158-1161.
> [3]Roweis, Sam T. "EM algorithms for PCA and SPCA." Advances in neural information processing systems. 1998.

---

### Official Review · AnonReviewer3 · 2018-11-03
**Novel idea of learning rules for unsupervised learning, need more theory/evidences on what/why meta objectives are sufficient for learning the unsupervised learning rules**

**Rating:** 8
**Confidence:** 3

**Review:**

This work brings a novel meta-learning approach that learns unsupervised learning rules for learning representations across different modalities, datasets, input permutation, and neural network architectures. The meta-objectives consist of few shot learning scores from several supervised tasks. The idea of using meta-objectives to learn unsupervised representation learning is a very interesting idea.

Authors mentioned that the creation of an unsupervised update rule is treated as a transfer learning problem, and this work is focused on learning a learning algorithm as opposed to structures of feature extractors. Can you elaborate on what aspect of learning rules and why they can be transferable among different modalities and datasets? For this type of meta-learning to be successful, can you discuss the requirements on the type of meta-objectives? Besides saving computational cost, does using smaller input dimensions favor your method over reconstruction type of semi-supervised learning, e.g. VAE?

In the section "generalizing over datasets and domains", the accuracy of supervised methods and VAE method are very close. This indicates those datasets may not be ideal to evaluate semi-supervised training.

In the section "generalizing over network architectures", what is the corresponding supervised/VAE learning accuracy?

In the experimentation section, can you describe in more details how input permutations are conducted? Are they re-sampled for each training session for tasks? If the input permutations are not conducted, will the comparison between this method, supervised and VAE be different?

After reviewing the author response, I adjusted the rating up to focus more on novelty and less on polished results.

---

> ### Author Response · Authors · 2018-11-14
> **Thank you**
>
> Thank you for your thoughtful review! Comments below:
>
> "Can you elaborate on what aspect of learning rules and why they can be transferable among different modalities and datasets?"
> This is a hypothesis based on the observation that hand designed learning rules transfer across modalities and datasets. We structure our learning rule in such a way as to aid this generalization. The specifics are largely inspired by biological neural networks--for instance the use of a neuron-local learning rule, and by the challenges associated with making meta-training stable--for instance, the use of normalization in almost every part of the system was found to be necessary to prevent meta-training from diverging. A better understanding of what aspects of learned learning rules transfer across datasets is a fascinating question and definitely something we are pursuing in future work.
>
> "For this type of meta-learning to be successful, can you discuss the requirements on the type of meta-objectives?"
> In general, the meta-objective has to be easily tractable and have a well defined derivative with respect to the final layer (e.g. from backpropagation during meta-training). It should also reflect, as well as possible, performance on the eventual task. In our case, we wanted the base network to learn a representation in an unsupervised way which easily exposed class labels or other high level attributes, so we chose our meta-objective to reward few-shot learning performance using the unsupervised representation. In early experiments, we explored a number of variations on our eventual meta-objective (e.g. clustering and softmax regression). We found similar performance for these variants, and chose the meta-objective we describe in the paper (least squares) because we believed it to be the simplest.
>
> "Besides saving computational cost, does using smaller input dimensions favor your method over reconstruction type of semi-supervised learning, e.g. VAE?"
> We only meta-train on the datasets with the smaller input size, but we test on both sizes (Figure 4). The VAE performance is comparable for the two input sizes, while the learned optimizer decreases in performance on mnist and remains constant on fashion mnist.
>
> "In the section "generalizing over network architectures", what is the corresponding supervised/VAE learning accuracy?"
> We have not run these experiments, but we would expect the performance of the VAE to go up with increased model size.
>
> "In the experimentation section, can you describe in more details how input permutations are conducted? Are they re-sampled for each training session for tasks? If the input permutations are not conducted, will the comparison between this method, supervised and VAE be different?"
> They are re-sampled for each new instantiation of an inner problem and kept constant while training that task. While we have not removed them, if we did we would expect the learned update rule to overfit to the meta-training distribution, causing improved performance on non-permuted image tasks, but extremely poor performance on permuted image tasks. Doing this would make comparisons to VAEs and supervised learning misleading however, as these two methods have no notion of spatial locality (whereas the learned optimizer now would). As a result, the learned optimizer’s relative performance would probably be a lot stronger. It would be very interesting in future work to use convnets for the base model--both for the learned update rule and the baselines. However, doing so would be a fairly involved process, requiring changes to the architecture of the learned update rule.

---

### Official Review · AnonReviewer1 · 2018-11-04
**an interesting approach to meta-learning, clear accept**

**Rating:** 8
**Confidence:** 4

**Review:**

This paper introduces a novel meta-learning approach to unsupervised representation learning where an update rule for a base model (i.e., an MLP) is meta-learned using a supervised meta-objective (i.e., a few-shot linear regression from the learned representation to classification GTs). Unlike previous approaches, it meta-learns an update rule by directly optimizing the utility of the unsupervised representation using the meta-objective. In the phase of unsupervised representation learning, the learned update rule is used for optimizing a base model without using any other base model objective. Experimental evaluations on few-shot classification demonstrate its generalization performance over different base architectures, datasets, and even domains.

+  Novel and interesting formulation of meta-learning by learning an unsupervised update rule for representation learning.
+  Technically sound, and well organized overall with details documented in appendixes.
+  Clearly written overall with helpful schematic illustrations and, in particular, a good survey of related work.
+ Good generalization performance over different (larger and deeper) base models, activation functions, datasets, and even a different modality (text classification).

-  Motivations are not very clear in some parts. E.g., the reason for learning backward weights (V), and the choice of meta-objective.
- Experimental evaluation is limited to few-shot classification, which is very close to the meta-learning objective used in this paper.
- The result of text classification is interesting, but not so informative given no further analysis. E.g., why domain mismatch does not occur in this case?

I enjoyed reading this paper, and happy to recommend it as a clear accept paper. The idea of meta-learning update networks looks a promising direction worth exploring, indeed.
I hope the authors to clarify the things I mentioned above. Experimental results are enough considering the space limit, but not great. Since the current evaluation task is quite similar to the meta-objective, evaluations on more diverse tasks would strengthen this paper.

Finally, this paper aims at unsupervised representation learning, but it’s not clear from the current title, which is somewhat misleading. I think that's quite an important feature of this paper, so I highly recommend the authors to consider a more informative title, e.g., `Learning Rules for Unsupervised Representation Learning’ or else.

---

> ### Author Response · Authors · 2018-11-14
> **Thank you**
>
> Thank you for your thoughtful review! Comments below:
>
> "Motivations are not very clear in some parts. E.g., the reason for learning backward weights (V), and the choice of meta-objective."
> Originally, we did not learn backward weights, but in an effort to make the learning rule more biologically inspired we removed the transposed weights in favor of learned backward weights [1]. In practice, performance is surprisingly quite similar with both versions.
>
> As per meta-objective: Exploring alternative meta-objectives would be very interesting! We choose the least squares meta-objective as it is allows us to compute the optimal final layer weights in closed form. This is important in that it allows us to easily differentiate the meta-objective with respect to the representation at the final layer (necessary for meta-training). We have explored alternative few-shot classification objectives (e.g. logistic regression, using implicit differentiation to get the appropriate derivative) but found performance to be similar and thus stuck with the simpler meta-objective.
>
> "Experimental evaluation is limited to few-shot classification, which is very close to the meta-learning objective used in this paper. "
> For simplicity, we used the same meta-objective at evaluation time. The use of different meta-objectives (at both meta-train and meta-test) is also very interesting to us and is something we would pursue in future work.
>
> "The result of text classification is interesting, but not so informative given no further analysis. E.g., why domain mismatch does not occur in this case?"
> Domain mismatch does occur--just later in meta-training. Because we are learning a learning rule, as opposed to features, we expect some generalization, after all, hand designed learning rules generalize across datasets. We get some transfer performance early in meta-training, but the meta-objective on text tasks diverges later in training. We will add a few sentences to this effect. Better understanding out of domain generalization is definitely of interest to us and we are pursuing it in future work.
>
> Paper Title: This is a good point and we plan to change the paper title to: "Meta-Learning Update Rules for Unsupervised Representation Learning".
>
>
> [1] Crick, F. The recent excitement about neural networks. Nature 337, 129–132 (1989).

---

### Public Comment · ~Deepak_Yadav1 · 2020-01-10
**Great paper | Possible incorrect pairing of variables names**

Hi,
Congratulations on your work!

I was going through your paper and on page 5, section 3.1 Base Model, it is written  x0, x1, x2, ....... xL as post non-linearity activations and z1, z2, z3 .....zL as pre non linearity activations.

I think it should be other way around. I am not confident on this, but I think this is a possible error on your part, or I might be interpreting things differently.

@Authors: Please let me know your thoughts on this edit.

Thanks!

---

### Meta-Review · Area_Chair1 · 2018-12-13
**Well written paper with an interesting idea**

**Confidence:** 4
**Recommendation:** Accept (Oral)

**Metareview:**

The reviewers all agree that the idea is interesting, the writing clear and the experiments sufficient.

To improve the paper, the authors should consider better discussing their meta-objective and some of the algorithmic choices.